# Subject Level Differential Privacy with Hierarchical Gradient Averaging

**Virendra J. Marathe**
Oracle Labs
virendra.marathe@oracle.com

**Pallika Kanani**
Oracle Labs
pallika.kanani@oracle.com

**Daniel Peterson**
Oracle Labs
daniel.peterson@oracle.com

## Abstract

Subject Level Differential Privacy (DP) is a granularity of privacy recently studied in the Federated Learning (FL) setting, where a subject is defined as an individual whose private data is embodied by multiple data records that may be distributed across a multitude of federation users. This granularity is distinct from item level and user level privacy appearing in the literature. Prior work on subject level privacy in FL focuses on algorithms that are derivatives of group DP or enforce user level *Local DP (LDP)*. In this paper, we present a new algorithm – *Hierarchical Gradient Averaging (*HiGradAvgDP*)* – that achieves subject level DP by constraining the effect of individual subjects on the federated model. We prove the privacy guarantee for *HiGradAvgDP* and empirically demonstrate its effectiveness in preserving model utility on the FEMNIST and Shakespeare datasets. We also report, for the first time, a unique problem of privacy loss composition, which we call *horizontal composition*, that is relevant only to subject level DP in FL. We show how horizontal composition can adversely affect model utility by either increasing the noise necessary to achieve the DP guarantee, or by constraining the amount of training done on the model.

## 1 Introduction

Cross-silo Federated Learning (FL) comprises a collection of institutions collaborating to train a common Machine Learning (ML) model [13]. This allows these institutions (also called *silos*) to jointly train the common model without sharing their private training data. Interestingly, a silo's private training data can be made up of private data of *individuals* (e.g. health records of patients of a hospital that is participating in a FL federation). In such instances, preservation of privacy of these individuals (also termed *data subjects*) is of paramount importance. However, the private data of a subject may be embodied by multiple data records that could be distributed across a multitude of federation silos.

As an example, consider a patient visiting different hospitals for treatment of different ailments. Each hospital contains multiple data records forming the patient's health history. These hospitals may decide to participate in a federation that uses their respective patients' health history records in their training datasets. Thus potentially distinct health history records of the same data subject can appear in the datasets of multiple hospitals. In the end, it is the privacy of these subjects that we want to preserve in a FL federation. The notion of subject level privacy perfectly captures this condition [19].

36th Conference on Neural Information Processing Systems (NeurIPS 2022).

Subject level Differential Privacy (DP) is a recently introduced granularity of privacy enforcement proposed in the FL setting [19]. This privacy guarantee is distinct from previously studied granularities of *item level* privacy [1, 9] and *user level* privacy [20], both of which may make sense in cross-device FL settings. Prior proposed algorithms [19] to enforce subject level DP either leverage the notion of group DP [10] or user-level Local DP (LDP) [8, 14, 30]. However, these algorithms incur significant model utility degradation, as we demonstrate in Section 5, since they amplify the noise to the contribution of individual subjects in the parameter updates.

In this paper, we present a novel algorithm, called *Hierarchical Gradient Averaging (*HiGradAvgDP*)*, that enforces subject level DP in the FL setting. The key insight behind *HiGradAvgDP* is that instead of amplifying the noise signal to match a subject's contribution to parameter updates in a training minibatch, we can *constrain* a subject's contribution to match a single data record's contribution to achieve subject level DP with less noise. We formally show that *HiGradAvgDP* leads to lower model utility loss for convex loss functions. We furthermore verify these insights empirically using the FEMNIST and Shakespeare datasets [6]. We also report an interesting observation about privacy loss composition for subject level DP in the FL setting. Specifically, we show that subject privacy loss composes across federation users in a FL training round. We call this *horizontal composition* of privacy loss and show that it can either lead to increase in the noise needed to enforce subject level DP, or can constrain the amount of training done on the model. Further exploration of solutions to the horizontal composition problem is a topic for future work.

The rest of the paper is organized as follows: We revisit the formal definition of subject level DP in Section 2. Section 3 briefly overviews existing algorithms for subject level DP in FL and then details our *HiGradAvgDP* algorithm, with formal arguments for its privacy guarantee. We present our empirical evaluation in Section 5 and conclude in Section 6.

## 2 Subject Level Differential Privacy

We first recap the definition of Differential Privacy [9]. Informally, DP bounds the maximum impact a single data item can have on the output of a randomized algorithm $\mathcal{A}$. Formally,

**Definition 2.1.** *A randomized algorithm* $\mathcal{A} : \mathcal{D} \to \mathcal{R}$ *is said to be* $(\varepsilon,\delta)$*-differentially private if for any two* adjacent *datasets* $D, D' \in \mathcal{D}$, *and set* $R \subseteq \mathcal{R}$,

$$\mathcal{P}(\mathcal{A}(D) \in R) \leq e^{\varepsilon} \mathcal{P}(\mathcal{A}(D') \in R) + \delta \tag{1}$$

*where* $D, D'$ *are adjacent to each other if they differ from each other by a single data item.* $\delta$ *is the probability of failure to enforce the* $\varepsilon$ *privacy loss bound.*

The above definition of DP can be easily recast in terms of data subjects to define subject level DP. Let $\mathcal{S}$ be the set of subjects whose data is hosted by the federation's users $\mathcal{U}$. Our definition of subject level DP is based on the observation that, even though the data of individual subjects $s \in \mathcal{S}$ may be physically scattered across multiple users in $\mathcal{U}$, the aggregate data across $\mathcal{U}$ can be logically divided into its subjects in $\mathcal{S}$ (i.e. $\mathcal{D}_{\mathcal{U}} = \bigcup_{s \in \mathcal{S}} \mathcal{D}_s$).

**Definition 2.2.** *Given a FL training algorithm* $\mathcal{F} : \mathcal{D}_{\mathcal{U}} \to \mathcal{M}$, *we say that* $\mathcal{F}$ *is subject level* $(\varepsilon, \delta)$*-differentially private if for any two adjacent subject sets* $S, S' \subseteq \mathcal{S}$, *and* $R \subseteq \mathcal{M}$,

$$\mathcal{P}(\mathcal{F}(\mathcal{D}_S) \in R) \leq e^{\varepsilon} \mathcal{P}(\mathcal{F}(\mathcal{D}_{S'}) \in R) + \delta \tag{2}$$

*where* $S$ *and* $S'$ *are adjacent subject sets if they differ from each other by a single subject.*

## 3 Subject Level Differential Privacy with Hierarchical Gradient Averaging

We assume a federation that contains a federation server that is responsible for (i) initialization and distribution of the model architecture to the federation users, (ii) coordination of training rounds, (iii) aggregation and application of model updates coming from different users in each training round, and (iv) redistribution of the updated model back to the users. Each federation user (i) receives updated models from the federation server, (ii) retrains the received models using its private training data, and (iii) returns updated model parameters to the federation server.

We assume that the federation users and the server behave as *honest-but-curious* participants in the federation: They do not interfere with or manipulate the distributed training process in any way, but

may be interested in analyzing received model updates. Federation users do not trust each other or the federation server, and must locally enforce privacy guarantees for their private data.

*HiGradAvgDP* is based on a federated version of the *DP-SGD* algorithm by Abadi et al. [1]. *DP-SGD* was originially not designed for FL, but can be easily extended to enforce item level DP in FL: The federation server samples a random set of users for each training round and sends them a request to perform local training. Each user trains the model locally using *DP-SGD*. Formally, the parameter update at step $t$ in *DP-SGD* can be summarized in the following equation:

$$\Theta_t = \Theta_{t-1} + \frac{\eta}{|B|} \left( \sum_{b \in B} \nabla \mathcal{L}_b^C (\Theta_{t-1}) + \mathcal{N}(0, C^2 \sigma^2) \right) \tag{3}$$

where, $B$ is the sampled minibatch of data items, $\nabla \mathcal{L}_b^C$ is the loss function's gradient, for data item $b$, clipped by the threshold $C$, $\sigma$ is the noise scale calculated using the moments accountant method, $\mathcal{N}$ is the Gaussian distribution used to calculate noise, and $\eta$ is the learning rate. Note that, in each sampled mini-batch $B$, the gradient for each data item is computed and clipped separately to limit the influence (*sensitivity*) of each data item on the loss function's gradient.

The users ship back updated model parameters to the federation server, which averages the updates received from all the sampled users. The server redistributes the updated model and triggers another training round if needed. The original paper [1] also proposed the moments accountant method for tighter composition of privacy loss bounds compared to prior work on strong composition [11].

### 3.1 Alternate Algorithms for Subject Level Differential Privacy

Prior work by Marathe and Kanani [19] presents two different algorithms for subject level DP enforcement in FL. Both algorithms are variants of the DP-SGD algorithm in FL as described above. The first algorithm is called *LocalGroupDP*, which uses the notion of *group differential privacy* [10] to enforce subject level DP. The intuition here is that a subject's contribution to gradients in a mini-batch can be trivially mapped to a group of data items sampled in the minibatch, thus making group DP a natural solution for subject level DP. The second algorithm is *UserLDP* that enforces a *stronger* user level *Local Differential Privacy (LDP)* [8, 14, 30] guarantee that amplifies the notion of a group to the entire sampled minibatch. More detailed description of both algorithms and their privacy guarantee formalism appears in Appendix A and Appendix B.

### 3.2 Hierarchical Gradient Averaging (*HiGradAvgDP*)

The key challenge to enforce subject level DP is that the following constraint seems fundamental: *To guarantee subject level DP, any training algorithm must obfuscate the entire contribution made by any subject in the model's parameter updates. UserLDP* complies with this constraint by scaling up the noise to the entire user signal, whereas *LocalGroupDP* complies with the constraint by scaling up the noise to a subject's signal in every minibatch. While *LocalGroupDP* may seem like an attractive alternative to *UserLDP*, the former's utility penalty due to group DP can be significant – the privacy loss $\varepsilon$ incurred due to group DP increases linearly with the group size, and the failure probability $\delta$ increases exponentially [10, 19]. For instance, even a group of size 2 effectively *halves* the available privacy budget $\varepsilon$ for training.

Our new algorithm, called *HiGradAvgDP* (Algorithm 1), takes a diametrically opposite view to comply with the same constraint: Instead of scaling the noise to a subject's group size (as is done in *LocalGroupDP*), *HiGradAvgDP scales down* each subject's mini-batch gradient contribution to the clipping threshold $C$. This is done in three steps: (i) collect data items belonging to a common subject in the sampled mini-batch, (ii) compute and clip gradients using the threshold $C$ for each individual data item of the subject, and (iii) average those clipped gradients for the subject, denoted by $g(\mathcal{S}_a^S)$. Clipping and then averaging gradients ensures that the entire subject's gradient norm is bounded by $C$. Subsequently, *HiGradAvgDP* sums all the per-subject averaged gradients along with the Gaussian noise, which are then averaged over the mini-batch size $B$. *HiGradAvgDP* gets its name from this average-of-averages step.

The Gaussian noise scale $\sigma$ is calculated independently at each user $u_i$ using standard parameters – the privacy budget $\varepsilon$, the failure probability $\delta$, total number of mini-batches $T.R$, and the sampling fraction per mini-batch $\frac{B}{|D_i|}$. The calculation uses the moments accountant method to compute $\sigma$.

**Algorithm 1:** Pseudo code for *HiGradAvgDP* that guarantees *subject level* DP via hierarchical gradient averaging.

**Parameters:** Set of $n$ users $\mathcal{U} = u_i, u_2, ..., u_n$; $\mathcal{D}_i$, the dataset of user $u_i$; $M$, the model to be trained; $\theta$, the parameters of model $M$; gradient norm bound $C$; noise scale $\sigma$; sample of users $U_s$; mini-batch size $B$; $R$ training rounds; $T$ batches per round; $\eta$ the learning rate; $\mathcal{S}_a^S$ the subset of data items from set $S$ that have $a$ as their subject.

```
 1  HiGradAvgDP(uᵢ):                           16  Server Loop:
 2    for t = 1 to T do                         17    for r = 1 to R do
 3      S = random sample of B data items       18      Uₛ = sample s users from 𝒰
          from 𝒟ᵢ                               19      for uᵢ ∈ Uₛ do
 4      for a ∈ subjects(S) do                  20        θᵢ = HiGradAvgDP(uᵢ)
 5        for sᵢ ∈ 𝒮ₐˢ do                       21      θ = (1/s) Σᵢ θᵢ
 6          Compute gradients:                  22      Send M to all users in 𝒰
 7          g(sᵢ) = ∇ℒ(θ, sᵢ)
 8          Clip gradients:
 9          ḡ(sᵢ) = Clip(g(sᵢ), C)
10        Average subject a's gradients:
11        g(𝒮ₐˢ) = (1/|𝒮ₐˢ|)(Σᵢ ḡ(sᵢ))
12      g̃ₛ = (1/B)(Σₐ∈subjects(S) g(𝒮ₐˢ) +
          𝒩(0, σ²C²I))
13      θ = θ − ηg̃ₛ
14    return M
15
```

To formally prove that *HiGradAvgDP* enforces subject level DP, we first provide a formal definition of *subject sensitivity* in a sampled mini-batch.

**Definition 3.1** (Subject Sensitivity). *Given a model $\mathcal{M}$, and a sampled mini-batch $S$ of training data, we define* subject sensitivity $\mathbb{S}^S$ *for $S$ as the maximum difference caused by any single subject $a \in subjects(S)$ in $\mathcal{M}$'s parameter gradients computed over $S$.*

**Lemma 3.1.** *For every sampled mini-batch $S$ in a sampled user $u_i$'s training round in* HiGradAvgDP, *the subject sensitivity $\mathbb{S}^S$ for $S$ is bounded by $C$; i.e. $\mathbb{S}^S \leq |C|$.*

**Theorem 3.2.** HiGradAvgDP *locally enforces subject level ($\varepsilon$,$\delta$)-differential privacy.*

Proofs for Lemma 3.1 and Theorem 3.2 appear in Appendix D. Moreover, due to space constraints we leave the formal analysis on utility loss bounds of *HiGradAvgDP*, as well as *UserLDP* and *LocalGroupDP*, to Appendix C.

### 3.3 Composition Over Multiple Training Rounds

Composition of privacy loss across multiple training rounds can be done by straightforward application of DP composition results, such as the moments accountant method that we use in our work. Thus the privacy loss $\varepsilon_r$ incurred in any single training round $r$ amplifies by a factor of $\sqrt{R}$ when federated training runs for $R$ rounds. We note that privacy losses are incurred by federation users independently of other federation users. Foreknowledge of the number of training rounds $R$ lets us calculate the Gaussian noise distribution's standard deviation $\sigma$ for a privacy loss budget of ($\varepsilon$,$\delta$) for the aggregate training over $R$ rounds.

## 4 Composing Subject Level DP Across Federation Users

At the beginning of a training round $r$, each sampled user receives a copy of the global model, with parameters $\Theta_{r-1}$, which it then retrains using its private data. Since all sampled users start retraining from the same model $\mathcal{M}_{\Theta_{r-1}}$, and independently retrain the model using their respective private data, parallel composition of privacy loss across these sampled users may seem to apply

naturally [21]. In that case, the aggregate privacy loss incurred across multiple federation users, via an aggregation such as federated averaging, remains identical to the privacy loss $\varepsilon_r$ incurred individually at each user. However, parallel composition was proposed for item level privacy, where an item belongs to at most one participant. With subject level privacy, a subject's data items can span across multiple users, which limits application of parallel privacy loss composition to only those federations where each subject's data is restricted to at most one federation user. In the more general case, we show that subject level privacy loss composes *sequentially* via the federated averaging aggregation algorithm used in our FL training algorithms.

Formally, consider a FL training algorithm $\mathcal{F} = (\mathcal{F}_l, \mathcal{F}_g)$, where $\mathcal{F}_l$ is the user local component, and $\mathcal{F}_g$ the global aggregation component of $\mathcal{F}$. Given a federation user $u_i$, let $\mathcal{F}_l : (\mathcal{M}, D_{u_i}) \rightarrow P_{u_i}$, where $\mathcal{M}$ is a model, $D_{u_i}$ is the private dataset of user $u_i$, and $P_{u_i}$ is the updated parameters produced by $\mathcal{F}_l$. Let $\mathcal{F}_g = \frac{1}{n} \sum_i P_{u_i}$, a parameter update averaging algorithm over a set of $n$ federation users $u_i$.

**Theorem 4.1.** *Given a FL training algorithm $\mathcal{F} = (\mathcal{F}_l, \mathcal{F}_g)$, in the most general case where a subject's data resides in the private datasets of multiple federation users $u_i$, the aggregation algorithm $\mathcal{F}_g$ sequentially composes subject level privacy losses incurred by $\mathcal{F}_l$ at each federation user.*

We term this sequential composition of privacy loss across federation users as *horizontal composition*. Horizontal composition has a significant effect on the number of federated training rounds permitted under a given privacy loss budget.

**Theorem 4.2.** *Consider a FL training algorithm $\mathcal{F} = (\mathcal{F}_l, \mathcal{F}_g)$ that samples $s$ users per training round, and trains the model $\mathcal{M}$ for $R$ rounds. Let $\mathcal{F}_l$ at each participating user, over the aggregate of $R$ training rounds, locally enforce subject level $(\varepsilon, \delta)$-DP. Then $\mathcal{F}$ globally enforces the same subject level $(\varepsilon, \delta)$-DP guarantee by training for $\frac{R}{\sqrt{s}}$ rounds.*

The main intuition behind Theorem 4.2 is that the $s$-way horizontal composition via $\mathcal{F}_g$ results in an increase in training mini-batches by a factor of $s$. As a result, the privacy loss calculated by the moments accountant method amplifies by a factor of $\sqrt{s}$, thereby forcing a reduction in number of training rounds by a factor of $\sqrt{s}$ to counteract the privacy loss amplification. This reduction in training rounds can have a significant impact on the resulting model's performance, as we demonstrate in Section 5. Proofs for Theorem 4.1 and Theorem 4.2 appear in Appendix D.

An alternate approach to account for horizontal composition of privacy loss is to simply scale the number of training minibatches (called *lots* by Abadi et al. [1]) by the number of federation users sampled in each training round. The scaled minibatch (lot) count can be used by each user to privately calculate the noise scale $\sigma$ at the beginning of the entire federated training process. An increase in the number of total minibatches does lead to a significant increase in the noise introduced in each minibatch's gradients, resulting in model performance degradation.

## 5 Empirical Evaluation

We implemented *UserLDP*, *LocalGroupDP*, and *HiGradAvgDP*, and a version of the DP-SGD algorithm by Abadi et al. [1] that enforces item level DP in the FL setting (*LocalItemDP*). We also compare these algorithms with a FL training algorithm, *FedAvg* [16], that does not enforce any privacy guarantees. All our algorithms are implemented in our distributed FL framework built on distributed PyTorch.

We focus our evaluation on Cross-Silo FL [13], containing 16 federation users (silos), which we believe is the most appropriate setting for the subject level privacy problem. We use the FEMNIST and Shakespeare datasets [6] for our evaluation. Due to space constraints, we report simple test accuracy results here (Figure 1) and the more exhaustive evaluation appears in Appendix E.

In FEMNIST, the hand-written numbers and letters can be divided based on authors, which ordinarily serve as federation users in FL experiments by most researchers. In Shakespeare, each character in the Shakespeare plays serves as a federation user. In our experiments however, the FEMNIST authors and Shakespeare play characters are treated as data subjects. To emulate the cross-silo FL setting, we report evaluation on a 16-user federation.

We use the CNN model on FEMNIST appearing in the LEAF benchmark suite [6] as our target model to train. More specifically, the model consists of two convolution layers interleaved with

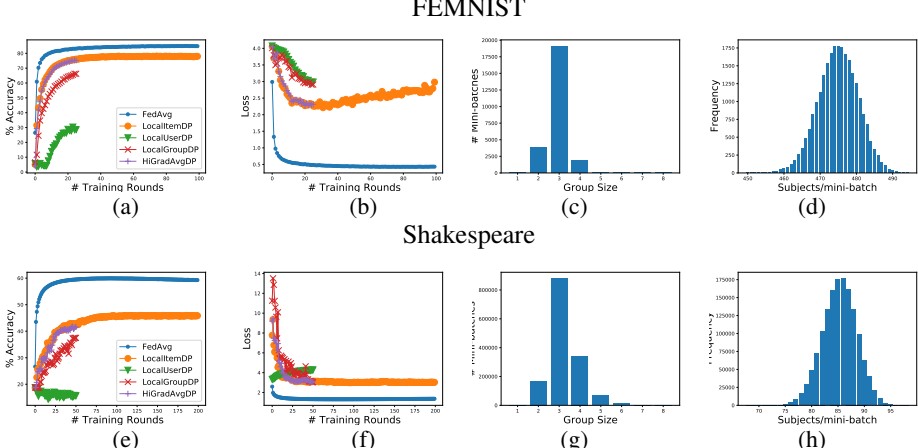

**Figure 1:** Average test accuracy and loss on the FEMNIST (a),(b) and Shakespeare (e),(f) datasets over training rounds for various algorithms. For DP guarantees: $\epsilon = 4.0$ and $\delta = 10^{-5}$ budgeted over 100 and 200 training rounds for FEMNIST and Shakespeare respectively. Model performance for the subject level privacy algorithms is constrained by the limited number of training rounds (25 for FEMNIST, and 50 for Shakespeare) permitted under the prescribed privacy budget. Number of mini-batches with subject group sizes over the entire training run for FEMNIST (c) and Shakespeare (g). Number of mini-batches with distinct subjects per mini-batch for FEMNIST (d) and Shakespeare (h). Minibatch size is 512 and 100 for FEMNIST and Shakespeare respectively.

ReLU activations and maxpooling, followed by two fully connected layers before a final log softmax layer. For the Shakespeare dataset we use a stacked LSTM model with two linear layers at the end.

In our implementations of *UserLDP*, *LocalGroupDP*, and *HiGradAvgDP*, we used the privacy loss horizontal composition accounting technique that reduces the number of training rounds by $\sqrt{s}$, where $s$ is the number of sampled users per training round. We experimented with the alternative approach that scales up the number of minibatches by $s$ to calculate a larger noise scale $\sigma$, but this approach consistently yielded worse model utility than our first approach. Hence here we report only the performance of our first approach.

Figure 1 shows performance of the models trained using the algorithms. *FedAvg* performs the best since it does not incur any DP enforcement penalties. Item level privacy enforcement in *LocalItemDP* results in performance degradation of 8% for FEMNIST and 22% for Shakespeare. The utility cost of user level LDP in *UserLDP* is quite clear from the figure. This cost is also reflected in the relatively high observed loss for the respective model. *LocalGroupDP* performs significantly better than *UserLDP*, but worse than *LocalItemDP*, by 15% on FEMNIST, and 18% on Shakespeare. The reason for *LocalGroupDP*'s worse performance is clear from Figure 1(c) and (g): the group size for a mini-batch tends to be dominated by 3 on both FEMNIST and Shakespeare, which cuts the privacy budget for these mini-batches by a factor of 3, leading to greater Gaussian noise, which in turn leads to model performance degradation.

*HiGradAvgDP* performs competitively with *LocalItemDP* for the 25 and 50 rounds it is trained for on FEMNIST and Shakespeare respectively. Figure 1 (d) and (h) show that instances of sampling multiple data items corresponding to the same subject in a single mini-batch are relatively low – the number of distinct subjects sampled per mini-batch of 512 for FEMNIST averages to 475, and per mini-batch of 100 for Shakespeare averages to 86. As a result, *HiGradAvgDP* incurs insignificant performance degradation for both datasets. However, the training round restriction does result in degradation of the final model produced by *HiGradAvgDP* compared to *LocalItemDP*: For FEMNIST, *HiGradAvgDP* gives 75.24% prediction accuracy after 25 rounds compared to 77.96% accuracy after 100 rounds with *LocalItemDP*. For Shakespeare, *HiGradAvgDP* gives 41.58% model accuracy after 50 rounds compared to 45.91% accuracy with *LocalItemDP* after 200 rounds.

## 6 Conclusion

In this paper we presented a novel algorithm called *Hierarchical Gradient Averaging (*HiGradAvgDP*)* that enforces the recently studied subject level DP guarantee in the FL setting [19]. We showed that the approach taken by *HiGradAvgDP* – scale a subject's signal down to a single data item's signal – instead of prior work's approach to scale the noise to a subject's signal, leads to much better model utility. We also studied the novel problem of *horizontal composition* of privacy loss for subjects in the FL setting, which can further degrade model utility. We leave mitigation of this utility degradation to future research.

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

---

**Algorithm 2:** Pseudo code for *UserLDP*.

---

**Parameters:** Set of $n$ users $\mathcal{U} = u_i, u_2, ..., u_n$; $\mathcal{D}_i$, the dataset of user $u_i$; $M$, the model to be trained; $\theta$, the parameters of model $M$; noise scale $\sigma$; gradient norm bound $C$; mini-batch size $B$; $R$ training rounds; the learning rate $\eta$.

---

| | | | |
|---|---|---|---|
| 1 | *UserLDP($u_i$):* | 12 | *Server Loop:* |
| 2 | **for** $t = 1$ **to** $T$ **do** | 13 | **for** $r = 1$ **to** $R$ **do** |
| 3 | $\quad$ $S$ = random sample of $B$ data items from $\mathcal{D}_i$ | 14 | $\quad$ $U_s$ = sample $s$ users |
| 4 | $\quad$ **Compute gradients:** | 15 | $\quad\quad$ from $\mathcal{U}$ |
| 5 | $\quad$ $g(S) = \triangledown\mathcal{L}(\theta, S)$ | 16 | $\quad$ **for** $u_i \in U_s$ **do** |
| 6 | $\quad$ **Clip gradients:** | 17 | $\quad\quad$ $\theta_i = UserDPSGD(u_i)$ |
| 7 | $\quad$ $\bar{g}(S) = g(S)/max(1, \frac{\|g(S)\|_2}{C})$ | 18 | $\quad$ $\theta = \frac{1}{s}\sum_{i=1}^{s}\theta_i$ |
| 8 | $\quad$ **Add Gaussian noise:** | 19 | $\quad$ Send $M$ to all users |
| 9 | $\quad$ $\tilde{g}(S) = \bar{g}(S) + \mathcal{N}(0, \sigma^2 C^2 \mathbf{I})$ | 20 | $\quad\quad$ in $\mathcal{U}$ |
| 10 | $\quad$ $\theta = \theta - \eta\tilde{g}(S)$ | | |
| 11 | **return** $\theta$ | | |

---

## A  User Level Local Differential Privacy (*UserLDP*)

In general, user level privacy [20] does not guarantee subject level privacy. However, we observe that a stronger privacy guarantee, called *Local Differential Privacy (LDP)* [8, 14, 30], enforced at user granularity, is sufficient to guarantee subject level privacy. There are strong parallels between the traditional LDP setting, where a data analyst can get access to the data only after it has been perturbed, and privacy in the FL setting, where the federation server gets access to parameter updates from users after they have been locally perturbed by the users. In fact, in the FL setting [27], LDP is a much stronger privacy guarantee than item level or user level DP in that it obfuscates the entire signal from a user to the extent that an adversary, even the federation server, cannot tell the difference between the signals coming from any two different users.

**Definition A.1.** *We say that FL algorithm $\mathcal{F}_l : \mathcal{D}_\mathcal{U} \to \mathcal{M}$ is user level $(\varepsilon,\delta)$-locally differentially private, where $\mathcal{D}_\mathcal{U}$ is the dataset domain of users in set $\mathcal{U}$, and $\mathcal{M}$ is the model parameter domain, if for any two users $u_1, u_2 \in \mathcal{U}$, and $S \subseteq \mathcal{M}$,*

$$\mathcal{P}(\mathcal{F}_l(D_{u_1}) \in S) \le e^\varepsilon \mathcal{P}(\mathcal{F}_l(D_{u_2}) \in S) + \delta \tag{4}$$

*where $D_{u_1}$ and $D_{u_2}$ are the datasets of users $u_1$ and $u_2$ respectively.*

User level LDP is a stronger privacy guarantee than subject level DP. More formally,

**Theorem A.1.** *User level $(\varepsilon,\delta)$-local differential privacy entails $(\varepsilon, \delta)$-subject level differential privacy.*

*Proof.* The definition of user level local DP can be easily reframed as a group differential privacy instance, where groups are as large as the entire dataset of each user. More specifically, Equation 4 is the definition of $(\varepsilon, \delta)$-group differential privacy for groups of size $g = max(|D_{u_1}|, |D_{u_2}|)$.

Beyond this observation, a simple application of Theorem B.3 proves $(\varepsilon, \delta)$-subject level differential privacy. $\square$

We now present a new user level $(\varepsilon,\delta)$-LDP algorithm called *UserLDP*. The underlying intuition behind this algorithm is to let the user locally inject enough noise to make its entire signal *indistinguishable* from any other user's signal. In every training round, each federation user enforces user level LDP independently of the federation and any other users in the federation. The federation server simply averages parameter updates received from users and broadcasts the new averaged parameters back to the users.

*UserLDP*'s pseudo code appears in Algorithm 2. *UserLDP* appears significantly similar to *DP-SGD*. The key difference is that while *DP-SGD* computes noise proportional to the gradient contribution of any single data item in a mini-batch, *UserLDP* computes noise proportional to the gradient contribution of the entire mini-batch, thus obfuscating the entire signal from the mini-batch. To guarantee DP, we need to first cap the sensitivity of each user $u_i$'s contribution to parameter updates. To that end, we focus on change affected by any mini-batch $b$ trained at $u_i$.

**Lemma A.2.** *For every mini-batch $b$ of a sampled user $u_i$'s training round in* UserLDP*, the sensitivity $\mathbb{S}_b$ of the computed parameter gradient is bounded by $C$; i.e. $\mathbb{S}_b \leq |C|$.*

*Proof.* The gradient clipping step of *UserDPSGD()* forces the bound on the gradient: $\|g(S)\|_2 \leq C$. Thus the sensitivity of parameter gradient $\mathbb{S}_b \leq |C|$. $\qquad\square$

**Theorem A.3.** UserLDP *enforces user level ($\varepsilon$,$\delta$)-local differential privacy for each mini-batch $b$.*

*Proof.* For any two users $u_1$ and $u_2$, the parameter gradient for each mini-batch is identically bounded by $C$. The Gaussian noise added per mini-batch parameter gradient is also drawn from the same distribution $\mathcal{N}(0, \sigma^2 C^2 \mathbf{I})$ scaled to the gradient bound $C$.

Over $T$ mini-batches, the cumulative parameter gradient is bounded by $TC$. Thus the aggregate sensitivity of parameter gradients over a training round ($T$ mini-batches) for both $u_1$ and $u_2$ is $TC$.

For every training round, we use the moments accountant technique [1] to compute the correct noise scale $\sigma$ given a privacy budget of $\varepsilon$, and a DP failure probability of $\delta$.

The resulting parameter updates from both $u_1$ and $u_2$ received by the federation server in a training round are effectively *locally randomized* [14].

Parameter updates from both $u_1$ and $u_2$, as observed by the federation server, can each be broken down into a signal bounded by $|TC|$ (we can ignore the scaling factor of the learning rate $\eta$ since it is identical for all users), and the cumulative noise from the distribution $\mathcal{N}(0, T\sigma^2 C^2 \mathbf{I})$ (by linear composition of random variables with identical Gaussian distributions). More precisely, let $\mathcal{C}$ be the change in parameters affected by any user $u_i$. Then

$$|\mathcal{C}(u_i)| \leq \eta(|TC| + |\mathcal{N}(0, T\sigma^2 C^2 \mathbf{I})|) \tag{5}$$

Also note that a dataset containing just $u_1$ is adjacent to a dataset containing just $u_2$ since they differ from each other in just one data item. Thus we can apply the classic proof of ($\varepsilon$,$\delta$)-DP for the Gaussian mechanism [10] Theorem A.1 to show that

$$\mathcal{P}(\mathcal{F}(u_1, M) \in M_o) \leq e^\varepsilon \mathcal{P}(\mathcal{F}(u_2, M) \in M_o) + \delta \tag{6}$$

We can further extend this guarantee over multiple training rounds with a $\sigma$ computed correctly using the moments accountant DP-composition technique.

$$\qquad\square$$

Intuitively, the parameter updates $\mathcal{M}_{u_1}$ and $\mathcal{M}_{u_2}$ computed by any single mini-batch at users $u_1$ and $u_2$ respectively, can be viewed as locally randomized responses [30] that satisfy Equation 4. Applying standard DP composition results [1, 11] to Theorem A.3, and combining it with Theorem A.1 locally proves subject level DP guarantee for *UserLDP* over individual training rounds.

## B   Locally Enforced Group Level Differential Privacy (*LocalGroupDP*)

While user level LDP is a stronger guarantee than subject level privacy, LDP in general is known to induce excessive noise in the training process, leading to significant utility degradation in the trained model [8, 14]. Intuitively, user level LDP is guaranteeing privacy at a coarser granularity (user level) as opposed to granularity of individual data subjects. As a result, we need to find better alternatives that more precisely calibrate noise proportional to a data subject's influence on training. A direct method to attain that is by obfuscating the effects of the *group* of data items belonging to the same subject. We can apply formalism of *group differential privacy* [10] to achieve this group level obfuscation. Formally (from [10]),

**Theorem B.1.** *Any ($\varepsilon, \delta$)-differentially private randomized algorithm $\mathcal{A}$ is ($g\varepsilon, ge^{(g-1)\varepsilon}\delta$)-differentially private for groups of size $g$. That is, given two $g$-adjacent datasets $D$ and $D'$, and $R \in \mathcal{M}$, where $\mathcal{M}$ is the output space domain,*

$$\mathcal{P}(\mathcal{A}(D) \in R) \leq e^{g\varepsilon} \mathcal{P}(\mathcal{A}(D') \in R) + ge^{(g-1)\varepsilon}\delta \tag{7}$$

*where $D$ and $D'$ are $g$-adjacent if they differ from each other in $g$ data items.*

---

**Algorithm 3:** Pseudo code for *LocalGroupDP* that guarantees *subject level* DP via group DP enforcement.

---

**Parameters:** Set of $n$ users $\mathcal{U} = u_i, u_2, ..., u_n$; $\mathcal{D}_i$, the dataset of user $u_i$; $M$, the model to be trained; $\theta$, the parameters of model $M$; gradient norm bound $C$; sample of users $U_s$; mini-batch size $B$; $Z$, largest group size in a mini-batch, $\sigma_Z$, precomputed noise scale for group of size $Z$; $R$ training rounds; $T$ batches per round; the learning rate $\eta$.

---

1   *LocalGroupDP($u_i$):*
2    **for** $t = 1$ **to** $T$ **do**
3      S = random sample of $B$ data items from $\mathcal{D}_i$
4      **for** $s_i \in S$ **do**
5        **Compute gradients:**
6        $g(s_i) = \nabla \mathcal{L}(\theta, s_i)$
7        **Clip gradients:**
8        $\bar{g}(s_i) = Clip(g(s_i), C)$
9      $Z = LrgGrpCnt(S)$
10     $\tilde{g}_s = \frac{1}{B}(\sum_i \bar{g}(s_i) + \mathcal{N}(0, \sigma_Z^2 C^2 \mathbf{I}))$
11     $\theta = \theta - \eta \tilde{g}_s$
12    **return** $M$

13   *Server Loop:*
14    **for** $r = 1$ **to** $R$ **do**
15      $U_s$ = sample $s$ users from $\mathcal{U}$
16      **for** $u_i \in U_s$ **do**
17        $\theta_i = LocalGroupDP(u_i)$
18      $\theta = \frac{1}{s} \sum_i \theta_i$
19      Send $M$ to all users in $\mathcal{U}$

---

Clearly, group DP incurs a big linear penalty on the privacy loss $\varepsilon$, and an even bigger penalty in the failure probability ($ge^{(g-1)\varepsilon}\delta$). Nevertheless, if $g$ is restricted to a small value (e.g. 2) the group DP penalty may be acceptable.

Theorem B.1 is a bi-directional implication. So it can be restated as follows:

**Corollary B.2.** *Any $(\mathcal{E}, \Delta)$-group differentially private algorithm $\mathcal{A}$, for a group size of $g$, is $(\mathcal{E}/g, \Delta/(ge^{(g-1)\frac{\mathcal{E}}{g}})$-differentially private.*

In the FL setting, subject level DP immediately follows from group DP for every sampled mini-batch of data items at every federation user. Let $S$ be a sampled mini-batch of data items at a user $u_i$, and $\mathcal{M}$ be the domain space of the ML model being trained in the FL setting.

**Theorem B.3.** *Let training algorithm $\mathcal{A}_g : S \to \mathcal{M}$ be group differentially private for groups of size $g$, and $l$ be the largest number of data items belonging to any single subject in S. If $l \leq g$, then $\mathcal{A}_g$ is subject level differentially private.*

Composition of group DP guarantees over multiple mini-batches and training rounds also follows established DP composition results [1, 11, 22]. For instance, the moments accountant method by Abadi et al. [1] shows that given an $(\varepsilon, \delta)$-DP gradient computation for a single mini-batch, the full training algorithm, which consists of $T$ mini-batches and a mini-batch sampling fraction of $q$, is $(O(q\varepsilon\sqrt{T}), \delta)$-differentially private. Theorem B.1 implies that the same algorithm is $(O(gq\varepsilon\sqrt{T}), ge^{(g-1)\varepsilon}\delta)$-differentially private for a group of size $g$.

We now present our new FL training algorithm, *LocalGroupDP*, that guarantees group DP. We make a critical assumption in *LocalGroupDP*: Each user can determine the subject for any of its data items. Absent this assumption, the user may need to make the worst case assumption that all data items used to train the model belong to the same subject. On the other hand, these algorithms are *strictly local*, and do not require that the identity of the subjects be resolved across users.

*LocalGroupDP* ( Algorithm 3) enforces subject level privacy locally at each user. Like prior work [1, 20, 24], we enforce DP in *LocalGroupDP* by adding carefully calibrated Gaussian noise in each mini-batch's gradients. Each user clips gradients for each data item in a mini-batch to a clipping threshold $C$ prescribed by the federation server. The clipped gradients are subsequently averaged over the mini-batch. The clipping step bounds the *sensitivity* of each mini-batch's gradients to $C$.

To enforce group DP, *LocalGroupDP* also locally tracks the item count of the subject with the largest number of items in the sampled mini-batch. This count determines the *group size* needed to enforce group DP for that mini-batch. This group size, $Z$ in Algorithm 3, helps determine the noise scale $\sigma_Z$, given the target privacy parameters $(\mathcal{E}, \Delta)$ over the entire training round. More specifically, we use the moments accountant method and Corollary B.2 to calculate $\sigma$ for $\varepsilon = \mathcal{E}/Z$,

and $\delta = \Delta/(Ze^{(Z-1)\frac{\mathcal{E}}{Z}})$. Note that the value of $Z$ can vary between mini-batches, due to which we represent the noise scale as $\sigma_Z$ in the pseudo code. $\sigma_Z$ is computed using the moments accountant method. The rest of the parameters to calculate $\sigma_Z - \mathcal{E}$, $\Delta$, total number of mini-batches ($T.R$), and sampling fraction ($B/total\ dataset\ size$) – remain the same throughout the training process. *LocalGroupDP* enforces $(\mathcal{E}/Z, \Delta/(Ze^{(Z-1)\frac{\mathcal{E}}{Z}})$-differential privacy, which by Corollary B.2 implies $(\mathcal{E}, \Delta)$-group differential privacy, hence subject level DP by Theorem B.3.

## C  Utility Loss

Our formal analysis leverages a long line of former work on differentially private empirical risk minimization (ERM) [2, 3, 7, 8, 12, 15, 24, 25, 26, 28]. In particular, we extend the notation of and heavily base our formal analysis on work by Bassily et al. [2], applying it to subject level DP in general, with specializations for our individual algorithms.

Let $\mathcal{Z}$ denote the data domain, and $\mathcal{D}$ denote a data distribution over $\mathcal{Z}$. We assume a $L$-Lipschitz convex loss function $\ell : \mathbb{R}^d \times \mathcal{Z} \to \mathbb{R}$ that maps a parameter vector $\mathbf{w} \in W$, where $W \subset \mathbb{R}^d$ is a convex parameter space, and a data point $z \in \mathcal{Z}$, to a real value.

**Definition C.1** ($\alpha$-Uniform Stability [2, 5]). *Let $\alpha > 0$. A (randomized) algorithm $\mathcal{A} : \mathcal{Z}^n \to W$ is $\alpha$-uniformly stable (w.r.t loss $\ell : W \times \mathcal{Z} \to \mathbb{R}$) if for any pair $S, S' \in \mathcal{Z}^n$ differing in at most one data point, we have*

$$\sup_{z \in \mathcal{Z}} \mathbb{E}_{\mathcal{A}}[\ell(\mathcal{A}(S), z) - \ell(\mathcal{A}(S'), z)] \leq \alpha$$

**Definition C.2** ($(k, \alpha)$-Uniform Stability). *Let $\alpha > 0$. A (randomized) algorithm $\mathcal{A} : \mathcal{Z}^n \to W$ is said to be $(k, \alpha)$-uniformly stable (w.r.t loss $\ell : W \times \mathcal{Z} \to \mathbb{R}$) if for any pair $S, S' \in \mathcal{Z}^n$ differing in at most $k$ data points, we have*

$$\sup_{z \in \mathcal{Z}} \mathbb{E}_{\mathcal{A}}[\ell(\mathcal{A}(S), z) - \ell(\mathcal{A}(S'), z)] \leq k\alpha$$

We use $(k, \alpha)$-uniform stability to represent the effect of a data subject with cardinality $k$ in the dataset. Thus algorithm $\mathcal{A}$ is $(k, \frac{\beta}{k})$-uniformly stable if

$$\mathbb{E}_{\mathcal{A}}[\ell(\mathcal{A}(S_k), z) - \ell(\mathcal{A}(S_0), z)] \leq \beta$$

**Lemma C.1.** *A (randomized) algorithm $\mathcal{A} : \mathcal{Z}^n \to W$ is $(k, \frac{\beta}{k})$-uniformly stable iff it is $\frac{\beta}{k}$-uniformly stable.*

*Proof.* Consider sets $S_0, S_1, S_2, ..., S_k \subset \mathcal{Z}$ such that $S_i = S_{i-1} \cup \{x_i\}$, for all $1 \leq i \leq k$, where $x_i \in \mathcal{Z}$. In other words, $S_i$ contains a single additional data point than $S_{i-1}$.

Assume that $\mathcal{A} : \mathcal{Z}^n \to W$ is $(k, \frac{\beta}{k})$-uniformly stable. Then we have

$$\mathbb{E}_{\mathcal{A}}[\ell(\mathcal{A}(S_k), z) - \ell(\mathcal{A}(S_0), z)]$$
$$= \mathbb{E}_{\mathcal{A}}[\sum_{i=1}^{k}(\ell(\mathcal{A}(S_i), z) - \ell(\mathcal{A}(S_{i-1}), z))]$$
$$= \sum_{i=1}^{k} \mathbb{E}_{\mathcal{A}}[\ell(\mathcal{A}(S_i), z) - \ell(\mathcal{A}(S_{i-1}), z)]$$
$$\leq \beta$$

By i.i.d. and symmetry assumptions, we get $\forall i \in 1, 2, ..., k$

$$\mathbb{E}_{\mathcal{A}}[\ell(\mathcal{A}(S_i), z) - \ell(\mathcal{A}(S_{i-1}), z)] \leq \frac{\beta}{k}$$

For the other direction of the *iff* we use the same sets $S_0, S_1, S_2, ..., S_k$, and assume $\forall i \in 1, 2, ..., k$

$$\mathbb{E}_{\mathcal{A}}[\ell(\mathcal{A}(S_i), z) - \ell(\mathcal{A}(S_{i-1}), z)] \leq \frac{\beta}{k}$$

Hence,

$$\mathbb{E}_{\mathcal{A}}[\ell(\mathcal{A}(S_k), z) - \ell(\mathcal{A}(S_0), z)]$$

$$= \mathbb{E}_{\mathcal{A}}[\sum_{i=1}^{k}(\ell(\mathcal{A}(S_i), z) - \ell(\mathcal{A}(S_{i-1}), z))]$$

$$= \sum_{i=1}^{k}\mathbb{E}_{\mathcal{A}}[\ell(\mathcal{A}(S_i), z) - \ell(\mathcal{A}(S_{i-1}), z)]$$

$$\leq \beta$$

$\square$

## C.1 General Utility Loss for Subject Level Privacy

Given the parameter vector $\mathbf{w} \in W$, dataset $S = s_1, s_2, ..., s_n$, and loss function $\ell$, we define the *empirical loss* of $\mathbf{w}$ as $\hat{\mathcal{L}}(\mathbf{w}; S) \triangleq \frac{1}{n}\sum_{i=1}^{n} \ell(\mathbf{w}, s_i)$, and the *excess empirical loss* of $\mathbf{w}$ as $\Delta\hat{\mathcal{L}}(\mathbf{w}; S) \triangleq \hat{\mathcal{L}}(\mathbf{w}; S) - \min_{\tilde{\mathbf{w}} \in \mathbf{W}} \hat{\mathcal{L}}(\tilde{\mathbf{w}}; S)$. Similarly we define the *population loss* of $\mathbf{w} \in W$ w.r.t. loss $\ell$ and a distribution $\mathcal{D}$ over $\mathcal{Z}$ as $\mathcal{L}(\mathbf{w}; \mathcal{D}) \triangleq \mathbb{E}_{z \sim \mathcal{D}}[\ell(\mathbf{w}, z)]$. The excess population loss of $\mathbf{w}$ is defined as $\Delta\mathcal{L}(\mathbf{w}; \mathcal{D}) \triangleq \mathcal{L}(\mathbf{w}; \mathcal{D}) - \min_{\tilde{\mathbf{w}} \in W} \mathcal{L}(\tilde{\mathbf{w}}; \mathcal{D})$.

**Lemma C.2** (from [2]). *Let $\mathcal{A} : \mathcal{Z}^n \to W$ be a $\frac{\beta}{k}$-uniformly stable algorithm w.r.t. loss $\ell : W \times \mathcal{Z} \to \mathbb{R}$. Let $\mathcal{D}$ be any distribution over $\mathcal{Z}$, and let $S \sim \mathcal{D}^n$. Then,*

$$\mathbb{E}_{S \sim \mathcal{D}^n, \mathcal{A}}[\mathcal{L}(\mathcal{A}(S); \mathcal{D}) - \hat{\mathcal{L}}(\mathcal{A}(S); S)] \leq \frac{\beta}{k} \tag{8}$$

Let $\mathcal{A}$ be a $L$-Lipschitz convex function that uses dataset $S$ to generate an approximate minimizer $\hat{\mathbf{w}}_S \in W$ for $\mathcal{L}(.; \mathcal{D})$. Thus the accuracy of $\mathcal{A}$ is measured in terms of *expected* excess population loss

$$\Delta\mathcal{L}(\mathcal{A}; \mathcal{D}) \triangleq \mathbb{E}[\mathcal{L}(\hat{\mathbf{w}}_S; \mathcal{D}) - \min_{\mathbf{w} \in W} \mathcal{L}(\mathbf{w}; \mathcal{D})] \tag{9}$$

**Lemma C.3.** *Let $\mathcal{A}_{SDP}$ be a $L$-Lipschitz randomized algorithm that guarantees subject level $(\varepsilon, \delta)$-DP. Let $T$ be the number of training iterations, $m$ the minibatch size per training step, and $\eta$ the learning rate. Then, $\mathcal{A}_{SDP}$ is $(\kappa, \alpha)$-uniformly stable, where $\kappa$ is the expected number of data items for any subject $s_p$ appearing in $\mathcal{A}_{SDP}$'s training dataset, and $\alpha = L^2\frac{(T+1)\eta}{n}$.*

*Proof.* Consider dataset $S$ comprising data items of $n_s$ subjects $s_1, ..., s_{p-1}, s_p, s_{p+1}, ..., s_{n_s}$, and dataset $S'$ comprising data items of $n_s - 1$ subjects $s_1, ..., s_{p-1}, s_{p+1}, ..., s_{n_s}$; i.e. $S$ and $S'$ differ from each other by a single data subject $s_p$. Let number of data items per subject $|s_i| > 0$.

Let $\mathbf{w}_0, \mathbf{w}_1, ...\mathbf{w}_T$ and $\mathbf{w}'_0, \mathbf{w}'_1, ..., \mathbf{w}'_T$ be the parameter values of $\mathcal{A}_{SDP}$ corresponding to $T$ training steps taken over input datasets $S$ and $S'$ respectively. Let $\xi_t \triangleq \mathbf{w}_t - \mathbf{w}'_t$ for any $t \in [T]$.

Assume random sampling with replacement for a minibatch of data items. Let $r$ be the number of data items in a sampled minibatch of size $m$ that belong to subject $s_p$. Then, by the non-expansiveness property of the gradient update step, we have

$$\|\xi_{\tau+1}\| \leq \|\xi_\tau\| + 2L\eta\frac{r}{m}$$

Note that $r$ is a binomial random variable. Thus the expected value of $r$, i.e. $\mathbb{E}[r] = m\frac{\kappa}{n}$, where $\kappa = \mathbb{E}[|s_p|]$. Thus $\kappa$ depends on the underlying data distribution $\mathcal{D}$. For instance, if $\mathcal{D}$ is a uniform distribution, $\kappa = \frac{n}{n_s}$, where $n_s$ is the number subjects in $S$. Assuming $\|\xi_0\| = 0$, taking expectation and using the induction hypothesis, we get

$$\mathbb{E}[\|\xi_{\tau+1}\|] \leq 2L\frac{\eta(\tau+1)\kappa}{n}$$
$$= 2L\frac{\eta(\tau+1)\kappa}{n}$$

Now let $\bar{\mathbf{w}}_T = \sum_{t=1}^{T} \mathbf{w}_t$ and $\bar{\mathbf{w}}_T' = \sum_{t=1}^{T} \mathbf{w}_t'$. Since $\ell$ is $L$-Lipschitz, for every $z \in \mathcal{Z}$ we get

$$\mathbb{E}[\ell(\bar{\mathbf{w}}_T, z) - \ell(\bar{\mathbf{w}}_T', z)] = \mathbb{E}[\ell(\mathcal{A}_{SDP}(S), z) - \ell(\mathcal{A}_{SDP}(S'), z)]$$
$$\leq L.\mathbb{E}[\|\bar{\mathbf{w}}_T - \bar{\mathbf{w}}_T'\|]$$
$$\leq L\frac{1}{T} \sum_{t=1}^{T} \mathbb{E}[\|\xi_t\|]$$
$$\leq L\frac{1}{T} \frac{2L\eta\kappa}{n} \sum_{t=1}^{T} t$$
$$= L^2 \frac{(T+1)\eta\kappa}{n}$$

$\square$

Note that the above bound is a scaled version (by $\kappa$) of the recently shown bound for item level DP [2]. Thus, intuitively in our case, the smaller the number of data items per subject in a dataset, the closer our bound is to that of item level DP. Our bound is identical to the item level DP bound in the extreme case where each subject has just one data item in the dataset.

From Lemma C.3, Equation 8 and Equation 9, and substituting $k = \kappa$ and $\beta = L^2 \frac{(T+1)\eta\kappa}{n}$ in Equation 8, we get

**Theorem C.4.** *Let $\mathcal{A}_{SDP}$ be a L-Lipschitz randomized algorithm that guarantees subject level $(\varepsilon, \delta)$-DP. Then its excess population loss is bounded by*

$$\Delta\mathcal{L}(\mathcal{A}_{SDP}; \mathcal{D}) \leq$$
$$\mathbb{E}_{S \sim \mathcal{D}^n, \mathcal{A}_{SDP}}[\hat{\mathcal{L}}(\bar{\mathbf{w}}_T; S) - \min_{\mathbf{w} \in W} \mathcal{L}(\mathbf{w}; S)] + L^2 \frac{\eta(T+1)}{n}$$

Interestingly, the above inequality appears to be identical to the excess population loss bound of work by Bassily et al. on item level DP [2]. However, only the third RHS term is identical, and the first two RHS terms evaluate to different quantities for all of our algorithms as we show below.

## C.2 Utility Loss for *LocalGroupDP* and *UserLDP*

We now formally show how *LocalGroupDP* amplifies the Gaussian noise that factors directly into excess population loss $\Delta\mathcal{L}$.

**Lemma C.5.** *Let $W$ be the $M$-bounded convex parameter space for* LocalGroupDP*, and $S \in \mathcal{Z}^n$ be the input (training) dataset. Let $(\varepsilon, \delta)$ be the subject level DP parameters for* LocalGroupDP*, $q$ be the minibatch sampling ratio, and $d$ the model dimensionality. Then, for any $\eta > 0$, the excess empirical loss of* LocalGroupDP *is bounded by*

$$\mathbb{E}[\hat{\mathcal{L}}(\bar{\mathbf{w}}_T; S)] - \min_{\mathbf{w} \in W} \mathcal{L}(\mathbf{w}; S) \leq$$
$$\frac{M^2}{2\eta T} + \frac{\eta L^2}{2} + \eta d \frac{c^2 k^2 q^2}{\varepsilon^2} \left( T \log \frac{k e^{(k-1)\varepsilon/k}}{\delta} \right)$$

*Proof.* From the classic analysis of gradient descent on convex-Lipschitz functions [2, 23], we get

$$\mathbb{E}[\hat{\mathcal{L}}(\bar{\mathbf{w}}_T; S)] - \min_{\mathbf{w} \in W} \mathcal{L}(\mathbf{w}; S) \leq \frac{M^2}{2\eta T} + \frac{\eta L^2}{2} + \eta \sigma^2 d$$

where the last term on the right hand side of the inequality is the additional empirical error due to that privacy enforcing noise.

By Theorem 1 from [1], the term $\sigma$ is lower bounded by

$$\sigma \geq c \frac{q \sqrt{T \log(1/\delta)}}{\varepsilon} \tag{10}$$

for item level DP. Extending the bound to group level DP, for groups of size $k$, by substituting $(\varepsilon, \delta)$ with $(k\varepsilon, k e^{(k-1)\varepsilon} \delta)$ gives us

$$\sigma \geq c \frac{kq \sqrt{T \log(\frac{k e^{(k-1)\varepsilon/k}}{\delta})}}{\varepsilon}$$

We get the theorem's inequality by substituting $\sigma$ as above.

$\square$

Combining Lemma C.5 with Theorem C.4 gives us

**Theorem C.6.** *The excess population loss of $\mathcal{A}_{\text{LocalGroupDP}}$ is satisfied by*

$$\Delta \mathcal{L}(\mathcal{A}_{\text{LocalGroupDP}}; \mathcal{D}) \leq$$
$$\frac{M^2}{2\eta T} + \frac{\eta L^2}{2} + \eta d \frac{c^2 k^2 q^2}{\varepsilon^2} \left( T \log \frac{k e^{(k-1)\varepsilon/k}}{\delta} \right)$$
$$+ L^2 \frac{\eta(T+1)}{n}$$

Note that the noise term amplifies quadratically with group size $k$, which leads to rapid utility degradation with increasing group size. The excess population loss measure for *UserLDP* can be obtained by simply replacing the group size term $k$ to the size of the minibatch $m$, which clearly leads to significantly greater noise amplification. Formally,

**Theorem C.7.** *The excess population loss of $\mathcal{A}_{\text{UserLDP}}$ is satisfied by*

$$\Delta \mathcal{L}(\mathcal{A}_{\text{UserLDP}}; \mathcal{D}) \leq$$
$$\frac{M^2}{2\eta T} + \frac{\eta L^2}{2} + \eta d \frac{c^2 m^2 q^2}{\varepsilon^2} \left( T \log \frac{m e^{(m-1)\varepsilon/m}}{\delta} \right)$$
$$+ L^2 \frac{\eta(T+1)}{n}$$

## C.3  Utility Loss for *HiGradAvgDP*

Recall that unlike *LocalGroupDP*, *HiGradAvgDP* does not scale up the noise to the group size of a subject in a minibatch. It instead scales down the gradients of all data items of the subject to a single data item's gradient bounds (established by the clipping threshold). As a result, the noise amplification we showed for *LocalGroupDP* does not exist for *HiGradAvgDP*. However, scaling down the gradient signal of a subject does indeed affect *HiGradAvgDP*'s utility. To show the effect formally we go back to the classic analysis of gradient descent for convex-Lipschitz functions, Lemma 14.1 in [23].

Let $\mathbf{w}^* = \underset{\mathbf{w} \in W}{argmin}\ \mathcal{L}(\mathbf{w}, S)$. Given that $\hat{\mathcal{L}}$ is a convex $L$-Lipschitz function, from [23] we have

$$\mathbb{E}[\hat{\mathcal{L}}(\bar{\mathbf{w}}_T; S)] - \mathcal{L}(\mathbf{w}^*; S) \leq \frac{1}{T} \sum_{t=1}^{T} \langle \mathbf{w}_t - \mathbf{w}^*, \nabla \hat{\mathcal{L}}(\mathbf{w}_t) \rangle \tag{11}$$

Consider $\mathbf{w}_{t+1} = \mathbf{w}_t + \eta \mathbf{v}_t$, where $\mathbf{v}_t = \nabla \hat{\mathcal{L}}(\mathbf{w}_t)$, and $\eta$ is the learning rate.

**Lemma C.8.** *Consider algorithm* $\mathcal{A}_{\text{HiGradAvgDP}}^{-}$ *that performs the same steps as* HiGradAvgDP *except for the noise injection step (at line 12 of Algorithm 3). Let* $\hat{\mathcal{L}}_{\text{HiGradAvgDP}}^{-}(\mathbf{w}; S)$ *be the* $L$-*Lipschitz continuous empirical loss function, and* $W$ *be the* $M$-*bounded convex parameter space for* $\mathcal{A}_{\text{HiGradAvgDP}}^{-}$. *If* $k$ *is the expected number of data items per subject in a sampled minibatch, then*

$$\mathbb{E}[\hat{\mathcal{L}}_{\text{HiGradAvgDP}}^{-}(\bar{\mathbf{w}}_T; S)] - \mathcal{L}_{\text{HiGradAvgDP}}^{-}(\mathbf{w}^*; S) \leq \frac{kM^2}{2\eta T} + \frac{\eta L^2}{2k}$$

*Proof.* Consider

$$\begin{aligned}
\langle \mathbf{w}_t - \mathbf{w}^*, \mathbf{v}_t \rangle &= \frac{k}{\eta} \langle \mathbf{w}_t - \mathbf{w}^*, \frac{\eta}{k} \mathbf{v}_t \rangle \\
&= \frac{k}{2\eta} (-\|\mathbf{w}_t - \mathbf{w}^* - \frac{\eta}{k} \mathbf{v}_t\|^2 + \|\mathbf{w}_t - \mathbf{w}^*\|^2 + \frac{\eta^2}{k^2} \|\mathbf{v}_t\|^2) \\
&= \frac{k}{2\eta} (-\|\mathbf{w}_{t+1} - \mathbf{w}^*\|^2 + \|\mathbf{w}_t - \mathbf{w}^*\|^2) + \frac{\eta}{2k} \|\mathbf{v}_t\|^2
\end{aligned}$$

Summing the equality over $t$ and collapsing the first term on the right hand side gives us

$$\begin{aligned}
\sum_{t=1}^{T} \langle \mathbf{w}_t - \mathbf{w}^*, \mathbf{v}_t \rangle \\
&= \frac{k}{2\eta} (\|\mathbf{w}_1 - \mathbf{w}^*\|^2 - \|\mathbf{w}_{T+1} - \mathbf{w}^*\|^2) + \frac{\eta}{2k} \sum_{t=1}^{T} \|\mathbf{v}_t\|^2 \\
&\leq \frac{k}{2\eta} (\|\mathbf{w}_1 - \mathbf{w}^*\|^2) + \frac{\eta}{2k} \sum_{t=1}^{T} \|\mathbf{v}_t\|^2 \\
&= \frac{k}{2\eta} (\|\mathbf{w}^*\|^2) + \frac{\eta}{2k} \sum_{t=1}^{T} \|\mathbf{v}_t\|^2,
\end{aligned}$$

assuming $\mathbf{w}_1 = 0$. Since $W$ is $M$ bounded and $\hat{\mathcal{L}}_{HiGradAvgDP}^{-}$ is $L$-Lipschitz, combining the above with Equation 11, we get

$$\mathbb{E}[\hat{\mathcal{L}}_{HiGradAvgDP}^{-}(\bar{\mathbf{w}}_T; S)] - \mathcal{L}_{HiGradAvgDP}^{-}(\mathbf{w}^*; S) \leq \frac{kM^2}{2\eta T} + \frac{\eta L^2}{2k}$$

$\square$

Now reintroducing the noise in *HiGradAvgDP* (at line 12 in Algorithm 3), with $\hat{\mathcal{L}}_{HiGradAvgDP}(\mathbf{w}; S)$ as the $L$-Lipschitz continuous loss function of *HiGradAvgDP*, we get

$$\mathbb{E}[\hat{\mathcal{L}}_{HiGradAvgDP}(\bar{\mathbf{w}}_T; S)] - \hat{\mathcal{L}}_{HiGradAvgDP}(\mathbf{w}^*; S) \leq$$
$$\frac{kM^2}{2\eta T} + \frac{\eta L^2}{2k} + \eta\sigma^2 d,$$

where the last term of the right hand side is the additional empirical error due to the privacy enforcing noise [2]. Combining the above inequality with Theorem C.4 we get

**Theorem C.9.** *The excess population loss of $\mathcal{A}_{\text{HiGradAvgDP}}$ is satisfied by*

$$\Delta\mathcal{L}(\mathcal{A}_{\text{HiGradAvgDP}}; \mathcal{D}) \leq \frac{k^2 M^2 + \eta^2 T L^2}{2k\eta T} + \eta\sigma^2 d$$
$$+ L^2 \frac{\eta(T+1)}{n}$$

*where $\sigma$ takes a value satisfying Equation 10.*

Note that the noise term for *HiGradAvgDP* is identical to that of the *DP-SGD* algorithm's noise term that enforces item level privacy guarantee [1]. However, the first term on the right hand side of the inequality scales linearly with $k$, the expected number of data items per subject. Thus we should expect some utility loss compared to *DP-SGD* for $k > 1$. At $k = 1$ *HiGradAvgDP* performs identically to *DP-SGD*.

## D   Additional Proofs

*Proof for Lemma 3.1:* Clipping gradients for each data item belonging to subject $a \in S$ before averaging the clipped gradients over the $\mathcal{S}_a^S$ ensures that the averaged gradients' L2-norm is bounded by $C$. Hence $\mathbb{S}^S \leq |C|$. □

*Proof for Theorem 3.2:* Bounding subject sensitivity $\mathbb{S}^S \leq |C|$ and scaling the Gaussian noise to that sensitivity bound clearly results in $(\varepsilon, \delta)$-DP guarantee in gradient computation for each subject of each mini-batch $S$. The mini-batch wide averaging of gradients $\tilde{g}_S$ done using the mini-batch size $B$ is justified since the per subject gradients' average can be restated as aggregation of scaled down gradients for data items corresponding to a subject $a$; i.e. $g(\mathcal{S}_a^S) = \sum_i \frac{\tilde{g}(s_i)}{|\mathcal{S}_a^S|}$. This gives us $B$ distinct gradient quantities for the data items in the sampled mini-batch $S$, and averaging these quantities requires the term $B$ in the denominator of the expression that computes $\tilde{g}_S$. We can apply the moments accountant method for privacy budget composition to extend the subject level $(\varepsilon, \delta)$-DP guarantee over $T$ mini-batches in a training round, aggregated over $R$ training rounds. Thus *HiGradAvgDP* locally enforces subject level $(\varepsilon, \delta)$-differential privacy. □

*Proof for Theorem 4.1:* Assume two distinct users $u_1$ and $u_2$ in a federation that host private data items of subject $s$. Let $\varepsilon_1$ and $\varepsilon_2$ be the respective subject privacy losses incurred by the two users during a training round.

It is straightforward to see that, in the worst case, data items of $s$ at users $u_1$ and $u_2$ can affect disjoint parameters in $\mathcal{M}$. Thus parameter averaging done by $\mathcal{F}_g$ simply results in summation and scaling of these disjoint parameter updates. As a result, the privacy losses, $\varepsilon_1$ and $\varepsilon_2$ incurred by $u_1$ and $u_2$ respectively are retained to their entirety by $\mathcal{F}_g$. In other words, privacy losses incurred for subject $s$ at users $u_1$ and $u_2$ compose sequentially. □

*Proof for Theorem 4.2:* The proof of training round constraints on horizontal composition can be broken down into two cases: First, each user in the federation locally trains for exactly $T$ mini-batches per training round, with exactly the same mini-batch sampling fraction $q$. Since horizontal composition is equivalent to sequential composition in the worst case, the moments accountant

method shows us that the resulting algorithm will be $(O(q\varepsilon\sqrt{TRs}), \delta)$-differentially private. To compensate for the $\sqrt{s}$ factor scaling of the privacy loss, $\mathcal{F}$ can be executed for $\frac{R}{\sqrt{s}}$ training rounds, yielding a $(O(q\varepsilon\sqrt{TR}), \delta)$-differentially private algorithm.

In the second case, each user $u_i$ may train for a unique number of mini-batches per training round, with a unique mini-batch sampling fraction dictated by $u_i$'s private dataset. Let $T_1, T_2, ..., T_s$, and $q_1, q_2, ..., q_s$ be the number of mini-batches per training round and mini-batch sampling fraction for the sampled users $u_1, u_2, ..., u_s$ respectively.

All our algorithms 2, 3, and 1 locally enforce subject level $(O(q_i\varepsilon\sqrt{T_iR}),\delta)$-DP at each user $u_i$. Privacy enforcement is done independantly at each federation user $u_i$. Furthermore, note that the privacy loss is uniformly apportioned among training rounds. Let $\mathcal{E} = O(q_i\varepsilon\sqrt{T_iR})$. Note that $\mathcal{E}$ is identical for each user $u_i$ in the federation. Thus if $\mathcal{E}$ is the total privacy loss budget over $R$ training rounds, a sampled user incurs $\varepsilon_r = \mathcal{E}/R$ privacy loss in a single training round $r$. Similarly, each of the $s$ sampled users in round $r$ incurs identical privacy loss $\varepsilon_r$ despite having different mini-batches per training round $T_i$ and mini-batch sampling fractions $q_i$. As noted earlier, these privacy losses compose horizontally (sequentially) via $\mathcal{F}_g$ over $s$ users, leading to privacy loss amplification by a factor of $\sqrt{s}$ as per the moments accountant method. To compensate for this privacy loss amplification, $\mathcal{F}$ can be executed for $\frac{R}{\sqrt{s}}$ training rounds. $\qquad\square$

## E  Empirical Evaluation in More Detail

We implemented all our algorithms *UserLDP*, *LocalGroupDP*, and *HiGradAvgDP*, and a version of the DP-SGD algorithm by Abadi et al. [1] that enforces item level DP in the FL setting (*LocalItemDP*). We also compare these algorithms with a FL training algorithm, *FedAvg* [16], that does not enforce any privacy guarantees. All our algorithms are implemented in our distributed FL framework built on distributed PyTorch.

We focus our evaluation on Cross-Silo FL [13], which we believe is the most appropriate setting for the subject level privacy problem. We use the FEMNIST and Shakespeare datasets [6] for our evaluation. In FEMNIST, the hand-written numbers and letters can be divided based on authors, which ordinarily serve as federation users in FL experiments by most researchers. In Shakespeare, each character in the Shakespeare plays serves as a federation user. In our experiments however, the FEMNIST authors and Shakespeare play characters are treated as data subjects. To emulate the cross-silo FL setting, we report evaluation on a 16-user federation.

We use the CNN model on FEMNIST appearing in the LEAF benchmark suite [6] as our target model to train. More specifically, the model consists of two convolution layers interleaved with ReLU activations and maxpooling, followed by two fully connected layers before a final log softmax layer. For the Shakespeare dataset we use a stacked LSTM model with two linear layers at the end.

We use $80\%$ of the training data for training, and $20\%$ for validation. Test data comes separately in FEMNIST and Shakespeare. Training and testing was done on a local GPU cluster comprising $2$ nodes, each containing $8$ Nvidia Tesla V100 GPUs.

We extensively tuned the hyperparameters of mini-batch size $B$, number of training rounds $T$, gradient clipping threshold $C$, and learning rate $\eta$. The final hyperparameters for FEMNIST were: $B = 512$, $T = 100$, $C = 0.001$, and learning rates $\eta$ of $0.001$ and $0.01$ for the non-private and private FL algorithms respectively. Shakespeare hyperparameters were: $B = 100$, $T = 200$, $C = 0.00001$, and learning rates $\eta$ of $0.0002$ and $0.01$ for the non-private and private FL algorithms.

In our implementations of all our algorithms *UserLDP*, *LocalGroupDP*, and *HiGradAvgDP*, we used the privacy loss horizontal composition accounting technique that reduces the number of training rounds by $\sqrt{s}$, where $s$ is the number of sampled users per training round. We experimented with the alternative approach that scales up the number of minibatches by $s$ to calculate a larger noise scale $\sigma$, but this approach consistently yielded worse model utility than our first approach. Hence here we report only the performance of our first approach.

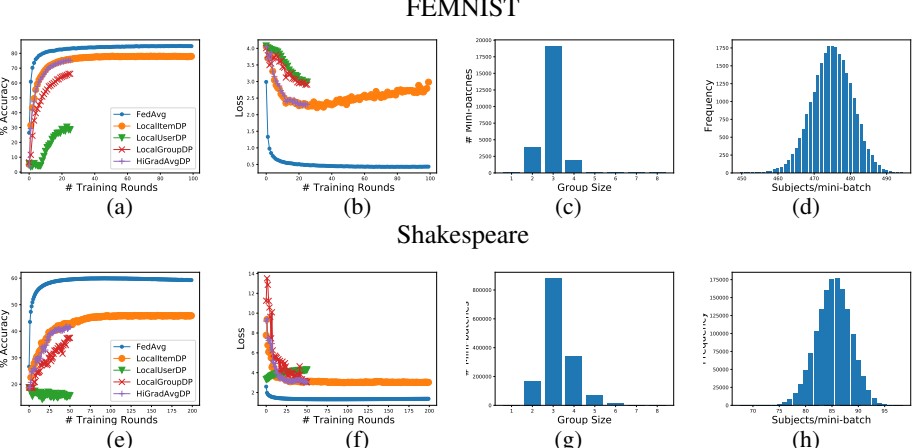

Figure 2: Average test accuracy and loss on the FEMNIST (a),(b) and Shakespeare (e),(f) datasets over training rounds for various algorithms. For DP guarantees: $\epsilon = 4.0$ and $\delta = 10^{-5}$ budgeted over all 100 and 200 training rounds for FEMNIST and Shakespeare respectively. Model performance for the subject level privacy algorithms is constrained by the limited number of training rounds (25 for FEMNIST, and 50 for Shakespeare) permitted under the prescribed privacy budget. Number of mini-batches with subject group sizes over the entire training run for FEMNIST (c) and Shakespeare (g). Number of mini-batches with distinct subjects per mini-batch for FEMNIST (d) and Shakespeare (h).

## E.1 FEMNIST and Shakespeare Performance

We first conduct an experiment that reports average test accuracy and loss at the end of each training round, over a total of 100 and 200 training rounds for FEMNIST and Shakespeare respectively. The FEMNIST dataset contains 3500 subjects, and the Shakespeare dataset contains 660 subjects. In both datasets each subject comprises hundreds of data items. Each subject's data items are uniformly distributed among the 16 federation users.

Figure 2 shows performance of the models trained using our algorithms. *FedAvg* performs the best since it does not incur any DP enforcement penalties. Item level privacy enforcement in *LocalItemDP* results in performance degradation of $8\%$ for FEMNIST and $22\%$ for Shakespeare. The utility cost of user level LDP in *UserLDP* is quite clear from the figure. This cost is also reflected in the relatively high observed loss for the respective model. *LocalGroupDP* performs significantly better than *UserLDP*, but worse than *LocalItemDP*, by $15\%$ on FEMNIST, and $18\%$ on Shakespeare. The reason for *LocalGroupDP*'s worse performance is clear from Figure 2(c) and (g): the group size for a mini-batch tends to be dominated by 3 on both FEMNIST and Shakespeare, which cuts the privacy budget for these mini-batches by a factor of 3, leading to greater Gaussian noise, which in turn leads to model performance degradation.

*HiGradAvgDP* performs competitively with *LocalItemDP* for the 25 and 50 rounds it is trained for on FEMNIST and Shakespeare respectively. Figure 2 (d) and (h) show that instances of sampling multiple data items corresponding to the same subject in a single mini-batch are relatively low – the number of distinct subjects sampled per mini-batch of 512 for FEMNIST averages to 475, and per mini-batch of 100 for Shakespeare averages to 86. As a result, *HiGradAvgDP* incurs insignificant performance degradation for both datasets. However, the training round restriction does result in degradation of the final model produced by *HiGradAvgDP* compared to *LocalItemDP*: For FEMNIST, *HiGradAvgDP* gives $75.24\%$ prediction accuracy after 25 rounds compared to $77.96\%$ accuracy after 100 rounds with *LocalItemDP*. For Shakespeare, *HiGradAvgDP* gives $41.58\%$ model accuracy after 50 rounds compared to $45.91\%$ accuracy with *LocalItemDP* after 200 rounds.

## E.2 Effect of Subject Data Distribution

While evaluation of our algorithms using a uniform distribution of subject data among federation users is a good starting point, often times the data distribution is non-uniform in real world settings. To emulate varying subject data distributions, we conduct experiments on the FEMNIST dataset where subject data is distributed among federation users according to the power distribution

$$P(x; \alpha) = \alpha x^{\alpha-1}, 0 \leq x \leq 1, \alpha > 0$$

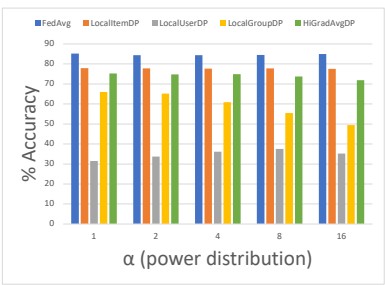

Figure 3: Model performance over FEMNIST dataset of our algorithms over different subject data distributions dictated by the parameter $\alpha$ of the power distribution.

Figure 3 shows performance of the models trained using our algorithms over varying subject data distributions of FEMNIST. As expected, different data distributions clearly do not significantly affect *FedAvg*, *LocalItemDP*, and *User-Local-SGD*. However, performance of the model trained using *LocalGroupDP* degrades substantially as the unevenness of data distribution increases, resulting in test accuracy under 50% for $\alpha = 16$. This degradation is singularly attributable to growth in subject group size per mini-batch – the average group size per mini-batch ranges from 3 when $\alpha = 2$ to 6 when $\alpha = 16$. This increase in group size significantly reduces the privacy budget leading to increase in Gaussian noise that restricts test accuracy. On the other hand, *HiGradAvgDP* appears to be much more resilient to non-uniform subject data distributions among federation users – test accuracy drops by just about 5% from $\alpha = 1$ (75.84% accuracy) to $\alpha = 16$ (71.89% accuracy). The corresponding subjects per minibatch observed in our experiments goes from an average of 475 to 310 respectively (not reported in detail due to space constraints).

### E.3    Small-FEMNIST Performance

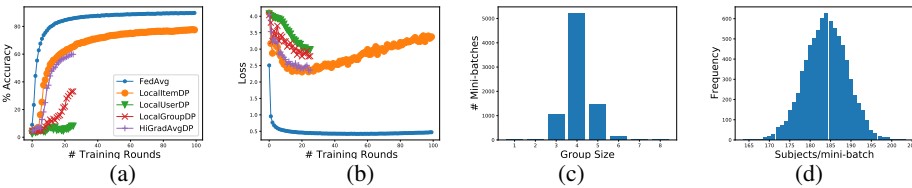

Figure 4: Average test accuracy (a) and loss (b) measured over training rounds for various algorithms on the Small-FEMNIST dataset (350 subjects) distributed among 16 federation users. For DP guarantees: $\epsilon = 4.0$ and $\delta = 10^{-5}$. Observed grouping sizes per mini-batch in *LocalGroupDP* aggregated across all mini-batches at all federation users in the training run (c), and number of subjects observed per mini-batch in *HiGradAvgDP* aggregated across all mini-batches at all federation users in the training run (d).

*HiGradAvgDP* appears to generally perform well when the number of subjects in the federation is sufficiently large. To study the effects of fewer subjects in a federation we experimented with a trimmed down version of FEMNIST, called Small-FEMNIST, that contains just the first 350 subjects in the aggregate dataset. For these experiments we reduced mini-batch size to 256. Figure 4 shows the performance of our algorithms on Small-FEMNIST. We note a substantial drop in the performance of *LocalGroupDP*, which can be explained by the increase in subject group size (to an average of 4) as can be seen in Figure 4 (c). There is also a noticeable drop in performance of *HiGradAvgDP*, which is attributable to a decrease in distinct subjects occuring per mini-batch (from Figure 4 (d)). This drop in number of subjects adversely affects performance of models trained using the algorithms that enforce subject level privacy.

