# OpenReview forum: "Subject Level Differential Privacy with Hierarchical Gradient Averaging"
_NeurIPS.cc/2022/Workshop/Federated_Learning — FL-NeurIPS 2022 Poster_

### Official Review · Reviewer_Thb7 · 2022-10-12

Summary of the paper:
This paper studies differentially private federated learning in a centralized setting, where different silos (e.g. hospitals) may share data from a same person. Key to this paper is the notion of ‘‘subject level differential privacy'', that formally takes this into account. The authors finally propose an algorithm -- HiGradAvgDP -- that combines user sampling, minibatch sampling, gradient clipping, noise injection. The authors take advantage of the subject level structure by ‘‘scaling up the noise to an entire subject signal'': a gaussian noise of variance \sigma^2 C^2 is injected for every subject in the considered minibatch. A general  \eps,\delta - DP analysis of the composition mechanisms at hand is performed, as well as an empirical evaluation.

Quality and clarity of the paper:
The paper is globally clearly written: the problem and objectives are well presented and formulated (what can be done better when subjects are present in different silos) the algorithm is clear.

Pros:
Originality and significance: the problem treated (DP federated learning, with this subject level formulation) is significant subject, and even if must admit that I am not quite familiar with this literature, I have no knowledge of works done in this area. In that sense, I believe that formulating this problem/scenario and providing the necessary definitions is in itself a contribution, especially suited for a workshop.
The provided algorithm is quite natural, the approach taken is well justified and seems to be justified empirically in the expermients provided.

Cons:
I however do not see any clear privacy-utility analysis of the proposed algorithm in the main text of the paper.

Overall, I range this submission as ''borderline accept'': the problem formulation, the writing, the algorithm counterweight the lack of privacy-utility analysis, the major weakness of this paper.
Still, I must assess that this paper is not in my area of expertise.


Minor typo: in (3) it should be a "-" and not a "+".

---

### Official Review · Reviewer_xBXi · 2022-10-18

Summary: This paper proposed a Hierarchical Gradient Averaging algorithm to guarantee the subject-level DP, which is different from the user-level or item-level DP in literature. The proposed algorithm achieves subject level DP by constraining the effect of individual subjects on the federated model. Experiments are conducted to validate the theory. And authors also provide an understanding of the privacy loss composition, horizontal composition, which is only related to subject level DP.

Strength:
This paper provides a new angle of understanding DP, with theoretical insight of subject DP. Theorem 4.1 and Theorem 4.2 are the main novelty of the paper, which defines and illustrates the horizontal composition.
The topic is widely studied in FL. DP guarantee analysis and algorithm design have many realistic applications and popular in the related research field. The experiments are well-conducted with details, and the results can validate the theory part.

Weakness:
It is very difficult to read and understand the paper. Many concepts in the paper are explained in words rather than mathematical equations, which causes confusion. For example, in Definition 3.1, subjective sensitivity is a key concept throughout the paper, but the concept is explained by “maximum difference”. What difference is that? It hasn’t been expressed in equation. And in Lemma 3.1, the definition “subject sensitivity” is not clear, but it gives a bound for it. Further, the Theorem 4.1 is totally described in words, and the sentence is not readable. It’s very difficult to understand the description. Since the paper causes confusion, I am not sure about the quality of the result unless the statements are clearly described in equations or explained in more details.
The proposed algorithm doesn’t seem new to me. As described in 3.2, the algorithm scales down each subject’s mini-batch gradient contribution to the clipping threshold C. This algorithm is very similar to UserLDP / LocalGroupDP  with some minor changes. So the algorithm basically has similar ideas to previous works.

---

### Decision · Program_Chairs · 2022-10-20

Accept (Poster)